# Sepsis recognition tools in acute ambulatory care: associations with process of care and clinical outcomes in a service evaluation of an Emergency Multidisciplinary Unit in Oxfordshire

Christian Fielder Camm,[1] Gail Hayward,[2] Tania C N Elias,[3,4] Jordan S T Bowen,[3,4] Roya Hassanzadeh,[5] Thomas Fanshawe,[2] Sarah T Pendlebury,[4,6,7] Daniel S Lasserson[4,8,9]

For numbered affiliations see end of article.

**Correspondence to**
Professor Daniel S Lasserson;
D.S.Lasserson@bham.ac.uk

## ABSTRACT

**Objective** To assess the performance of currently available sepsis recognition tools in patients referred to a community-based acute ambulatory care unit.

**Design** Service evaluation of consecutive patients over a 4-month period.

**Setting** Community-based acute ambulatory care unit.

**Data collection and outcome measures** Observations, blood results and outcome data were analysed from patients with a suspected infection. Clinical features at first assessment were used to populate sepsis recognition tools including: systemic inflammatory response syndrome (SIRS) criteria, National Early Warning Score (NEWS), quick Sequential Organ Failure Assessment (qSOFA) and National Institute for Health and Care Excellence (NICE) criteria. Scores were assessed against the clinical need for escalated care (use of intravenous antibiotics, fluids, ongoing ambulatory care or hospital treatment) and poor clinical outcome (all-cause mortality and readmission at 30 days after index assessment).

**Results** Of 533 patients (median age 81 years), 316 had suspected infection with 120 patients requiring care escalated beyond simple community care. SIRS had the highest positive predictive value (50.9%, 95% CI 41.6% to 60.3%) and negative predictive value (68.9%, 95% CI 62.6% to 75.3%) for the need for escalated care. Both NEWS and SIRS were better at predicting the need for escalated care than qSOFA and NICE criteria in patients with suspected infection (all P<0.001). While new-onset confusion predicted the need for escalated care for infection in patients ≥85 years old (n=114), 23.7% of patients ≥85 years had new-onset confusion without evidence for infection.

**Conclusions** Acute ambulatory care clinicians should use caution in applying the new NICE endorsed criteria for determining the need for intravenous therapy and hospital-based location of care. NICE criteria have poorer performance when compared against NEWS and SIRS and new-onset confusion was prevalent in patients aged ≥85 years without infection.

## Strengths and limitations of this study

► Acute ambulatory care is applicable to an increasingly prevalent healthcare delivery model.
► Consecutive patient evaluation to minimise selection bias.
► Not all elements of the sepsis recognition models were available for scoring.

## INTRODUCTION

Sepsis, defined as 'life-threatening organ dysfunction caused by dysregulated host response to infection',[1] has a high mortality[2] but early recognition allows for prompt and effective treatment.[3] Therefore, tools for accurate recognition are needed at the earliest possible point after patients attend an initial healthcare assessment setting.

The Third International Consensus Definitions for Sepsis and Septic Shock (SEPSIS-3) advocates the Sequential Organ Failure Assessment (SOFA)[1] score to evaluate sepsis. However, SOFA is not practical to use in the community and the quick SOFA (qSOFA) score has been developed for use in rapid triage and in settings without access to blood tests.[4] The National Institute for Health and Care Excellence (NICE) has recently introduced new guidance regarding sepsis, including a tool for the diagnosis and triage of patients with suspected sepsis in community settings.[2] The use of the qSOFA and the NICE high-risk and moderate-risk criteria (NICE-HR and NICE-MR, respectively) are a significant departure from the older systemic inflammatory response syndrome (SIRS) criteria.[5] In addition to sepsis recognition, the National Early Warning Score (NEWS)

has been advocated by the Royal College of Physicians to track the clinical condition of patients presenting acutely for hospital and community assessment.[6]

New tools for sepsis recognition have not been evaluated in all healthcare environments that provide initial clinical assessments. Lower sensitivity or specificity of either qSOFA or NICE criteria compared with existing tools for detecting sepsis at initial assessment, particularly in the community, could result in poorer care overall. Lower sensitivity results in missed cases of sepsis compared with current recognition tools and lower specificity results in the unnecessary escalation of care including rapid transfer from community settings to emergency departments, unnecessary use of broad-spectrum antibiotics and rapid fluid administration.

The performance of newer recognition scores, such as qSOFA and NICE criteria, may vary with clinical setting. Low specificity of NICE criteria has been suggested by the large number of acute medical unit admissions it detects as needing rapid assessment within 1 hour, in comparison with qSOFA and SIRS.[7 8] Furthermore, among emergency department patients with low qSOFA scores, a significant proportion require critical care interventions which would not have been anticipated using qSOFA[9] and the now abandoned SIRS criteria have shown more accurate prediction of death than qSOFA in some emergency department cohorts.[10] Given that in intensive care unit patients, SOFA has a better performance than SIRS,[11 12] it is likely that the clinical setting and nature of the patient cohorts in which these tools are applied have a strong influence on their accuracy to detect sepsis and stratify patients based on risk of poor outcomes.

The acute ambulatory care model, which is recommended as a strategy in acute medicine to meet increasing demands for care within constrained resources,[13] represents a particular challenge for the recognition of sepsis. While ambulatory care is becoming increasingly prevalent,[14] there are very few tools designed to support clinical decision making in the ambulatory care environment. Patients who are deemed suitable for ambulatory care do not have very disordered physiology,[15] even though there may be an emerging underlying significant disease process. Furthermore, identification of sepsis in older patients with frailty is complex as acute functional decline has many other causes,[16] yet mortality and morbidity related to sepsis is higher in the older population.[17] One of the critical decisions in ambulatory care management of suspected infection is selecting which patients can be safely discharged with oral therapy and follow-up from their general practitioner and which patients are at higher risk of clinical deterioration and require care that is enhanced, either through an ambulatory platform with daily parenteral antibiotics with senior clinical review or medical admission.

The ambulatory care clinician, faced with the diagnostic and management challenge of early sepsis recognition and the need to safely manage patients on an ambulatory pathway, can choose from four different sepsis recognition tools (SIRS, NEWS, NICE-HR and NICE MR and qSOFA). This study aims to evaluate the performance of these tools by examining their association with processes of care and clinical outcomes in patients with suspected infection in an acute ambulatory care setting. Given the uncertainty about application to older people in whom there is a wide differential diagnosis for acute frailty syndromes such as confusion, we undertook a subgroup analysis in patients 85 years or older.

## METHODS

The direct care team collected healthcare data as part of routinely provided healthcare from consecutive patients presenting to the Abingdon Hospital's Emergency Multi-disciplinary Unit (EMU) run by Oxford Health NHS Foundation Trust between 12/08/2015 and 08/12/2015. Patients were included if they were residents within Oxfordshire, allowing for appropriate follow-up details to be obtained and had a clinical phenotype suggesting possible infection, in keeping with the recommended application of sepsis screening tools.[5] The use of routine healthcare data collected by the direct care team for the purposes of determining adherence to national guidance and application of clinical prediction tools to inform future service development was prospectively approved as a service evaluation by the Oxford University Hospitals NHS Foundation Trust (providers of the medical care on EMU), with audit Datix reference number 3812, and permission to publish was granted by the Oxford Health NHS Foundation Trust R&D Department.

### Study setting

The EMU model of care provides assessment and treatment for adults with acute illness where an ambulatory treatment path is considered likely by the initial contact healthcare provider. Facilities include point-of-care blood testing (electrolytes, renal function, blood gases, lactate, C reactive protein, troponin, International Normalised Ratio (INR)), plain X-ray and intravenous delivery of medications and fluids. Referrals are primarily from general practitioners and paramedic teams in the community. Patients whose care needs cannot be met with an ambulatory pathway are admitted to either a community hospital or an associated acute hospital.

### Process of care and clinical outcomes

Details regarding the patients' presentation, investigations, treatment and outcome were prospectively documented at the time of assessment on the EMU. Patients' initial assessment was undertaken using a structured clinical clerking proforma including a brief cognitive test (the Abbreviated Mental Test Score and delirium screen) as described previously.[18] Data for analyses were extracted from the clerking proforma supplemented by information from ambulance sheets, general practitioner referral letters and communications with patients and relatives as listed in online supplementary table 1. Patients were

considered to have a possible infection if a clinical suspicion of infection was recorded by the treating clinician or if the presenting complaint suggested an infective cause for their symptoms. Processes of care included use of intravenous antibiotics, intravenous fluids and pathway of care (ambulatory vs hospital) and readmission within 30 days. Clinical outcome was 30-day all-cause mortality.

We tested the ability of sepsis scoring systems to select patients who require continued monitoring and treatment from the medical team, either via ambulatory or bed-based pathways. Patients were considered to have required escalated care for an infection if *all* of the following conditions were met: (1) the patient had a possible infection, (2) the patient received intravenous antibiotics and intravenous fluids, (3) the patient was either admitted to an inpatient facility or subsequently brought back to the EMU for ≥1 daily reviews, or enrolled in the 'hospital @ home' programme to receive ongoing intravenous antibiotics and daily review after initial EMU assessment. Furthermore, patients with suspected infection who did not meet conditions (2) and (3) were considered to have needed escalated care if they died within 7 days of review or required readmission for infection within 30 days. This strategy was used as it collectively identifies patients for whom initial discharge from ambulatory care with no or oral antibiotics may have been inappropriate.

### Sepsis scoring systems

We evaluated the accuracy of SIRS,[18] qSOFA,[6] NEWS,[2] NICE-HR and NICE-MR from NICE Guideline NG51[7] for detecting the need for escalated care to treat suspected bacterial infection. Each method was undertaken in accordance with guidance provided by issuing bodies. Features associated with these tools available in our data set and cut-off values for prediction of sepsis or poor outcome are detailed in online supplementary table 2. Within selected scoring systems (NICE-MR, NICE-HR), some variables are not systematically available at the first assessment in an acute setting (eg, urine output, usual blood pressure values for calculation of difference from observed value) and online supplementary table 2 lists these variables together with any additional clinical features that were not systematically sought within the clerking proforma. The scores were calculated without use of these items which would reflect real-world practice after implementation. A cut-off score of >4 was used for NEWS in line with guidance from the Royal College of Physicians (London) which details this threshold for separating low-risk patients from those at increased risk.[7] For cases in which individual items required for a score were missing, the score for that individual was calculated without that item, in line with clinical practice. During the analysis, NICE-MR and NICE-HR were combined in some analyses to replicate the use of the moderate criteria in clinical practice. We did not separately analyse the Sepsis Trust Red/Amber flag system (https://sepsistrust.org/education/clinical-tools/) as features of this system were incorporated into the new NICE guidelines.

### Statistical analysis

Continuous variables are expressed as either mean (SD) or median (IQR) and compared across suspected infection/no suspected infection and escalated care/no escalated care groups using the independent Student's t-test or Mann-Whitney U-test, respectively. Categorical and ordinal variables are reported as frequency (percentage) and compared between groups using the $\chi^2$ or Fisher's exact test. The diagnostic accuracy of each predictive tool was assessed using positive predictive value (PPV), negative predictive value (NPV), sensitivity and specificity, with 95% CIs. Predictive tools were compared with one another using McNemar's test, and we consider P values of less than 0.005 to be statistically significant in these comparisons (based on a conservative Bonferroni correction for 10 comparisons). The performance of sepsis tools was assessed in the whole data set and in a subgroup of the oldest patients (≥85 years) in predicting the need for escalated care, mortality and readmission within 30 days of index assessment. Analysis was performed using V.9 of the SAS system for Windows (Copyright © 2002–2010 by SAS Institute, Cary, North Carolina, USA).

### PATIENT INVOLVEMENT

This study was a service evaluation. There was no direct patient involvement in its conduct.

### RESULTS

A total of 533 patients were assessed during the observation time period. Ten patients were not local residents and did not have follow-up data and were excluded from the analysis (figure 1). Demographic data of the study population are provided in table 1. Three hundred and sixteen patients were considered to have a potential infection with 120 patients requiring escalated treatment for infection. Features most significantly associated with need for escalated care were increased heart rate (89.6 (18.9) vs 83.8 (17.8), P=0.006), increased temperature (37.0 (0.9) vs 36.6 (0.6), P<0.001) and increased white cell count (14.5 (17.4) vs 9.3 (3.7), P=0.003). In the whole cohort, there was no association between new-onset confusion and need for escalated care (22/120 (20.0%) vs 27/196 (14.0%), P=0.172). A full list of features associated with escalated care is displayed in table 1.

Of 186 patients ≥85 years old (45% of the cohort), 114 patients had a suspected infection as the cause for their presentation. Among patients ≥85 years with suspected infection, those who met criteria for escalated care (n=35, 30.7%) also demonstrated the positive findings for the whole cohort in terms of difference in physiological markers, but new confusion was significantly associated with escalated care in this subgroup (table 1).

The PPV, NPV, sensitivity and specificity of each assessment tool against the need for escalated care, 30-day mortality, 30-day admission and a composite marker of 30-day mortality or admission for the cohort with

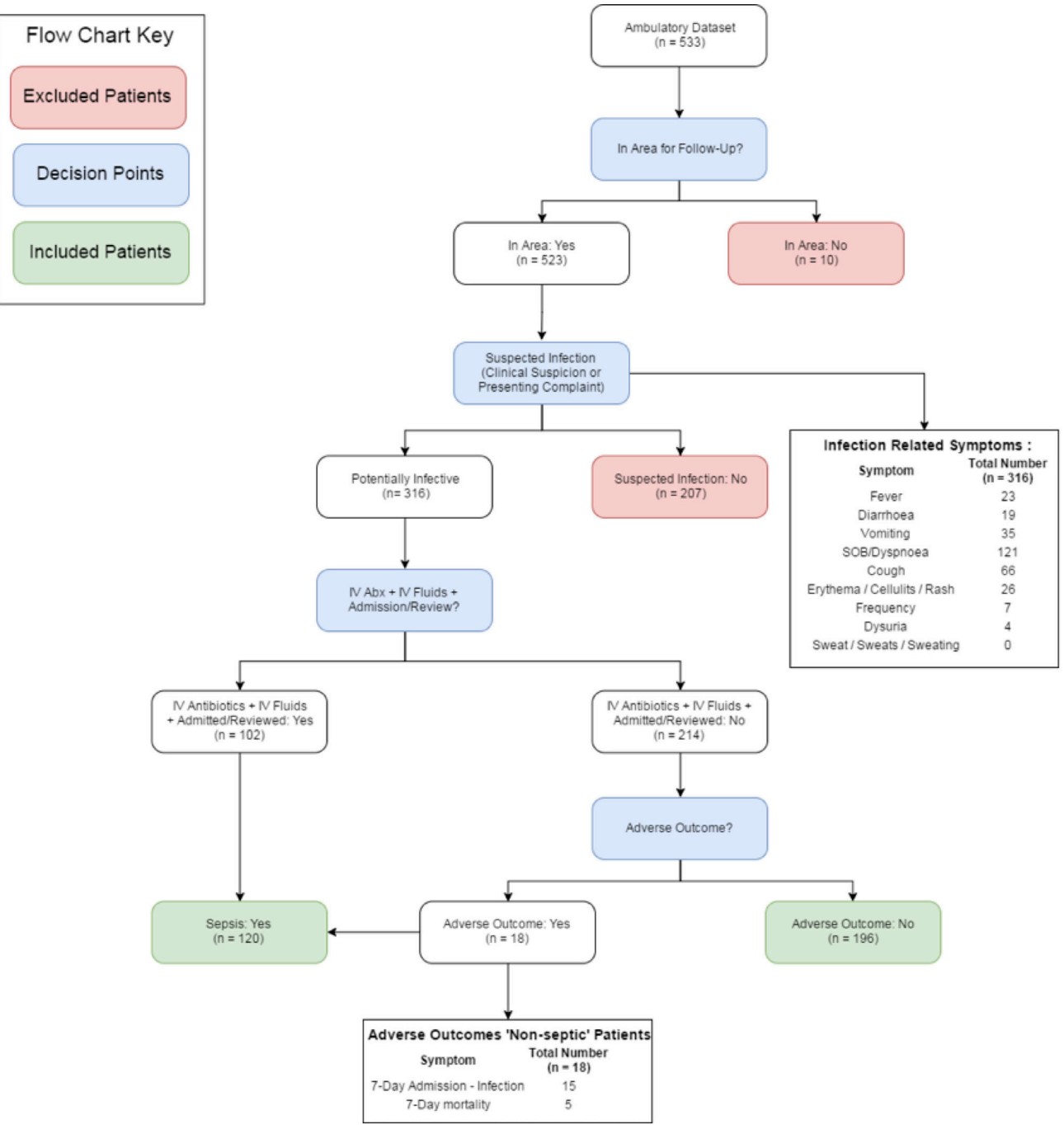

**Figure 1** Patient flow from clinical presentation to outcomes. IV, intravenous.

suspected infection are shown in table 2. SIRS had the highest PPV (50.9%, 95% CI 41.6% to 60.3%) and NPV (68.9%, 95% CI 62.6% to 75.3%). Using McNemar's test, the combined high-risk and moderate-risk NICE criteria were shown to be significantly worse at predicting escalated care compared with SIRS (P<0.001), NEWS (P<0.001) and NICE-HR criteria alone (P=0.004) (table 3), a result driven by very low specificity of the combined high-risk and moderate-risk NICE criteria (21.4%, 95% CI 15.7 to 27.2). Furthermore, both SIRS and NEWS were shown to have better performance against qSOFA (both P<0.001). NEWS and SIRS had

the highest specificity for detecting 30-day mortality (67.0, 95% CI 61.6 to 72.4 and 73.5, 95% CI 68.5 to 78.6, respectively).

Among patients ≥85 years old, the PPV and NPV of both SIRS and NEWS were higher than NICE or qSOFA scores for escalated care and for the combined 30-day clinical outcome (see online supplementary table 3). While NICE criteria had the highest sensitivity at detecting poor clinical outcomes (mortality or admission within 30 days) (88.4%), this was at the cost of very low specificity (14.3%). Receiver operating characteristic curves for the ability of the different sepsis scoring systems to identify

**Table 1** Demographic characteristics of patients assessed on the Emergency Multidisciplinary Unit during this analysis with suspected infection (n=316)

| | All patients with suspected infection | | | | ≥85 years old | | | | <85 years old | | | |
| | Need for escalated care | | | | Need for escalated care | | | | Need for escalated care | | | |
| Item | N | Yes | No | P values | N | Yes | No | P values | N | Yes | No | P values |
|---|---|---|---|---|---|---|---|---|---|---|---|---|
| Subjects | 316 | 120 (38.0%) | 196 (62.0%) | – | 114 | 35 (30.7%) | 79 (69.3%) | – | 202 | 85 (42.1%) | 117 (57.9%) | – |
| Proportion male | 316 | 57 (47.5%) | 79 (40.3%) | 0.210 | 114 | 10 (28.6%) | 33 (41.8%) | 0.180 | 202 | 47 (55.3%) | 46 (39.3%) | 0.024 |
| Age (years) | 316 | 74.5 (16.6) | 75.2 (18.3) | 0.708 | 114 | 90.2 (3.3) | 89.7 (3.9) | 0.442 | 202 | 68.0 (15.5) | 65.4 (17.7) | 0.283 |
| Heart rate (beats per minute) | 315 | 89.6 (18.9) | 83.8 (17.8) | 0.006 | 114 | 89.5 (19.2) | 79.29 (15.5) | 0.003 | 201 | 89.6 (18.9) | 86.9 (18.6) | 0.306 |
| Respiratory rate | 314 | 21.6 (6.4) | 21.3 (5.3) | 0.643 | 113 | 24.4 (8.2) | 20.7 (5.2) | 0.016 | 201 | 20.5 (5.1) | 21.7 (5.4) | 0.094 |
| Systolic blood pressure | 314 | 132.2 (20.1) | 135.9 (26.7) | 0.172 | 113 | 129.1 (21.4) | 132.0 (26.1) | 0.534 | 201 | 133.5 (19.5) | 138.5 (26.9) | 0.133 |
| Oxygen saturations (%) | 313 | 96 (94–98) | 97 (95–98) | 0.007 | 113 | 96 (93–97) | 97 (95–98) | 0.009 | 200 | 96 (94–98) | 97 (94–98) | 0.132 |
| Temperature (°C) | 314 | 37.0 (0.9) | 36.6 (0.6) | <0.001 | 113 | 36.9 (0.9) | 36.5 (0.6) | 0.044 | 201 | 37.1 (0.9) | 36.7 (0.6) | 0.001 |
| AMTS | 303 | 10 (8–10) | 10 (9–10) | 0.155 | 107 | 7 (5–9) | 9 (7–10) | 0.008 | 196 | 10 (9–10) | 10 (9–10) | 0.532 |
| Known dementia diagnosis | 316 | 21 (17.5%) | 23 (11.7%) | 0.151 | 114 | 9 (25.7%) | 14 (17.7%) | 0.327 | 202 | 12 (14.1%) | 9 (7.7%) | 0.140 |
| New confusion | 316 | 22 (18.3%) | 27 (13.8%) | 0.172 | 114 | 15 (42.9%) | 18 (22.8%) | 0.012 | 202 | 7 (8.2%) | 9 (7.7%) | 0.769 |
| White cell count (x10⁹/L) | 274 | 14.5 (17.4) | 9.3 (3.7) | 0.003 | 99 | 13.9 (6.4) | 9.3 (4.1) | 0.001 | 175 | 14.7 (20.4) | 9.3 (3.3) | 0.028 |
| Mortality (30 day) | 316 | 14 (11.7%) | 11 (5.6%) | 0.053 | 114 | 8 (22.9%) | 5 (6.3%) | 0.010 | 202 | 6 (7.1%) | 6 (5.1%) | 0.567 |
| Barthel score ≥15 | 316 | 80 (66.7%) | 150 (76.5%) | 0.056 | 114 | 17 (48.6%) | 53 (67.1%) | 0.061 | 202 | 63 (74.1%) | 97 (82.9%) | 0.129 |
| Sepsis scores at initial presentation | | | | | | | | | | | | |
| NEWS | 316 | 3 (1–5) | 3 (1–4) | | 114 | 4 (3–9) | 3 (1–4) | | 202 | 3 (1–5) | 3 (1–5) | |
| SIRS criteria | 316 | 1 (1–2) | 1 (0–2) | | 114 | 2 (1–2) | 1 (0–1) | | 202 | 1 (0–2) | 1 (0–2) | |
| qSOFA criteria | 316 | 1 (0–1) | 1 (0–1) | | 114 | 1 (1–2) | 1 (0–1) | | 202 | 0 (0–1) | 1 (0–1) | |
| NICE HR | 316 | 45 (37.5%) | 69 (35.2%) | | 114 | 21 (60.0%) | 31 (39.2%) | | 202 | 24 (28.2%) | 38 (32.5%) | |
| NICE MR and HR | 316 | 96 (80.0%) | 154 (78.6%) | | 114 | 31 (88.6%) | 69 (87.3%) | | 202 | 65 (76.5%) | 85 (72.7%) | |

Results displayed as N (%), mean (SD) or median (IQR) unless otherwise indicated. For the scoring systems evaluated, all data points evaluated were available in the following numbers: SIRS (n=272), NICE high risk (n=314), NICE moderate risk and high risk (n=314), qSOFA (n=314) and NEWS (n=313).
AMTS, Abbreviated Mental Test Score; NEWS, National Early Warning Score ; NICE, National Institute for Health and Care Excellence; qSOFA, quick Sequential Organ Failure Assessment; SIRS, systemic inflammatory response syndrome.

Table 2  Analysis of scoring systems versus outcome measures in patients presenting to the Emergency Multidisciplinary Unit with suspected infection (n=316)

| Outcome measure test | PPV (%) | NPV (%) | Sensitivity (%) | Specificity (%) |
|---|---|---|---|---|
| **Escalated care** | | | | |
| SIRS | 50.9 (41.6 to 60.3) | 68.9 (62.6 to 75.3) | 46.7 (37.7 to 55.6) | 72.4 (66.2 to 78.7) |
| NICE HR | 39.5 (30.5 to 48.4) | 62.9 (56.2 to 69.5) | 37.5 (28.8 to 46.2) | 64.8 (58.1 to 71.5) |
| NICE MR and HR | 38.4 (32.4 to 44.4) | 63.6 (52.0 to 75.2) | 80.0 (72.8 to 87.2) | 21.4 (15.7 to 27.2) |
| qSOFA | 37.2 (30.2 to 44.2) | 60.9 (52.6 to 69.2) | 56.7 (47.8 to 65.5) | 41.3 (34.4 to 48.2) |
| NEWS >4 | 48.9 (38.4 to 59.3) | 66.2 (60.1 to 72.4) | 35.8 (27.3 to 44.4) | 77.0 (71.2 to 82.9) |
| **30-Day mortality** | | | | |
| SIRS | 12.7 (6.5 to 19.0) | 94.7 (91.6 to 97.7) | 56.0 (36.5 to 75.5) | 67.0 (61.6 to 72.4) |
| NICE HR | 12.3 (6.3 to 18.3) | 94.6 (91.4 to 97.7) | 56.0 (36.5 to 75.5) | 65.6 (60.2 to 71.1) |
| NICE MR and HR | 9.6 (5.9 to 13.3) | 98.5 (95.5 to 100.0) | 96.0 (88.3 to 100.0) | 22.3 (17.6 to 27.1) |
| qSOFA | 11.5 (6.9 to 16.1) | 97.0 (94.1 to 99.9) | 84.0 (69.6 to 98.4) | 44.3 (38.6 to 50.0) |
| NEWS >4 | 12.5 (5.6 to 19.4) | 93.9 (90.7 to 97.0) | 44.0 (24.5 to 63.5) | 73.5 (68.5 to 78.6) |
| **30-Day readmission** | | | | |
| SIRS | 82.7 (75.7 to 89.8) | 34.0 (27.5 to 40.4) | 40.1 (33.7 to 46.5) | 78.7 (70.1 to 87.2) |
| NICE HR | 74.6 (66.6 to 82.6) | 29.7 (23.4 to 36.0) | 37.4 (31.1 to 43.7) | 67.4 (57.7 to 77.2) |
| NICE MR and HR | 74.8 (69.4 to 80.2) | 39.4 (27.6 to 51.2) | 82.4 (77.4 to 87.3) | 29.2 (19.8 to 38.7) |
| qSOFA | 74.9 (68.6 to 81.1) | 32.3 (24.4 to 40.3) | 60.4 (54.0 to 66.7) | 48.3 (37.9 to 58.7) |
| NEWS >4 | 78.4 (69.8 to 87.0) | 30.7 (24.7 to 36.7) | 30.4 (24.4 to 36.4) | 78.7 (70.1 to 87.2) |
| **30-Day composite** | | | | |
| SIRS | 83.6 (76.7 to 90.5) | 33.5 (27.1 to 39.9) | 40.2 (33.8 to 46.5) | 79.3 (70.8 to 87.8) |
| NICE HR | 75.4 (67.5 to 83.3) | 29.2 (22.9 to 35.5) | 37.6 (31.3 to 43.8) | 67.8 (58.0 to 77.6) |
| NICE MR and HR | 75.6 (70.3 to 80.9) | 39.4 (27.6 to 51.2) | 82.5 (77.6 to 87.5) | 29.9 (20.3 to 39.5) |
| qSOFA | 75.4 (69.2 to 81.6) | 31.6 (23.7 to 39.5) | 60.3 (53.9 to 66.6) | 48.3 (37.8 to 58.8) |
| NEWS >4 | 79.5 (71.1 to 88.0) | 30.3 (24.3 to 36.2) | 30.6 (24.6 to 36.5) | 79.3 (70.8 to 87.8) |

PPV, NPV, sensitivity and specificity show percentage and 95 % CI.
NEWS, National Early Warning Score; NICE, National Institute for Health and Care Excellence; NPV, negative predictive value; PPV, positive predictive value; qSOFA, quick Sequential Organ Failure Assessment; SIRS, systemic inflammatory response syndrome.

patients requiring escalated care are shown in online supplementary figure 1.

## DISCUSSION

Of the tools assessed, SIRS and NEWS were shown to have the highest PPV and NPV for escalated care in the data set overall and in the subgroup of patients ≥85 years. While NICE criteria and qSOFA had better performance in detecting requirement for escalated care in patients ≥85 years, their poorer specificity contributed to a lower PPV. The presence of new confusion was associated with escalated care among patients with suspected infection in the subgroup of patients aged ≥85 years.

Although SIRS was shown to be the most predictive of escalated care in patients with suspected infection in this study, the combined high-risk and moderate-risk NICE criteria would capture the most patients requiring escalated care but at the cost of very low specificity, that is, a high false-positive rate. These results contrast with the previously suggested limitations in predictive ability for poor outcomes when compared against other potential sepsis recognition and stratification tools which have led to a move away from using SIRS.[5] This may reflect applicability in different patient cohorts as Kaukonen et al demonstrated that 12.1% of intensive care unit patients with severe sepsis did not meet ≥2 SIRS criteria.[2 19 20] However, they also showed that those with SIRS positive sepsis had significantly increased mortality compared with those not meeting these criteria. Churpek et al demonstrated that 47% of an acute patient cohort met ≥2 SIRS criteria during admission and concluded that screening patients for sepsis based on these criteria would be unwise.[19] However, this paper did not select patients in whom infection was suspected and instead applied it to all admissions, irrespective of presenting complaint. A recent analysis from the same group has also suggested that in those admitted to hospital and suspected of infection, discrimination for in-hospital mortality was highest using

**Table 3** McNemar's analysis of each predictive tool compared against one another in determining the need for escalated care in patients with suspected infection (n=316)

| | NEWS >4 | qSOFA | NICE MR and HR | NICE HR |
|---|---|---|---|---|
| SIRS | 0.2 (P=0.893) | 21.3 (P<0.001) | 25.7 (P<0.001) | 6.63 (P=0.01) |
| NICE HR | 5.9* (P=0.034) | 5.9 (P=0.015) | 8.5 (P=0.004) | – |
| NICE MR and HR | 18.5* (P<0.001) | 1.27 (P=0.259) | – | – |
| qSOFA | 19.3* (P<0.001) | – | – | – |

Values represent row-listed diagnostic tool performed better than column-listed diagnostic tool.
*Represents column-listed diagnostic tool performed better than row-listed diagnostic tool.
A P value of <0.005 is considered significant following Bonferroni correction for multiple analysis.
NEWS, National Early Warning Score; NICE-MR and HR, combined analysis of National Institute of Health and Care Excellence moderate-risk and high-risk criteria; qSOFA, quick Sequential Organ Failure Assessment; SIRS, systemic inflammatory response syndrome

NEWS.[21] Our results add to this information by providing support for the use of NEWS in identifying patients in need of care escalation in a community setting.

The importance of identifying sepsis in the initial stages of clinical contact to allow for prompt and effective treatment has been well documented.[20] However, our results have shown that those systems endorsed by both NICE and Sepsis-3 have limitations related to high false-positive rates as compared with both SIRS and NEWS, which could lead to substantial overtreatment.

The introduction of new-onset confusion as a predictor of sepsis in NICE guidance risks increasing the use of inappropriate antibiotics in the older patient with frailty. Although our results show a significant association between the development of new confusion and the need for escalated care among patients ≥85 years with infection, there was a high prevalence of new-onset confusion in this age group without any evidence of underlying infection. While infection is a principle cause of delirium,[3] there is limited high-quality evidence that altered mental state in older adults is predictive of sepsis itself[22] and delirium has a wide range of potential causes in older people.[5]

Our data show that the new combined NICE-HR and NICE-MR criteria are likely to significantly increase the number of patients that are determined as potentially having sepsis at initial assessment in ambulatory care. As such, this is likely to increase the number of patients receiving intravenous antibiotics which may conflict with the national and global priorities of antibiotic stewardship.[22] Furthermore, in the ambulatory care setting their introduction may increase conversion to formal hospital admission which our data suggest will not be justified based on the clinical outcomes we recorded.

## Limitations

Several limitations of this work are acknowledged. Data collection for this work was undertaken at a time when SIRS was the recommended assessment tool for potential sepsis. As such, incorporation bias may explain some degree of the correlation between this method of assessment and the diagnosis and treatment of sepsis. However, SIRS was not calculated as part of the standard assessment proforma in this cohort and white cell count was not available at initial review as it is not part of the point-of-care testing portfolio. This limitation would not explain the correlation between SIRS and NEWS with mortality at 30 days.

Not all elements of the new NICE criteria for high-risk and moderate-risk sepsis were available for this analysis. However, the limited specificity of both these methods in determining sepsis would only be worsened by the addition of further criteria to this analysis. Furthermore, we argue that several of those unavailable elements (eg, mottled skin, cyanosis and urine output) represent those that would either not be available in the acute ambulatory setting or are based on subjective assessment and therefore more likely to vary between clinicians.

## CONCLUSIONS

This study demonstrates that new NICE-recommended sepsis tools do not outperform the existing tools in an ambulatory care setting in predicting the need for escalated care for infection and clinical outcomes at 30 days. Although NICE tools had higher sensitivity, this was at the expense of very low specificity. SIRS had the highest predictive values but the requirement of a white cell count implies that it can only be used widely if this is tested at the point of care. Further research should be undertaken to determine applicability of NICE sepsis guidance in different healthcare settings where acutely unwell patients are assessed before widespread adoption in the National Health Service.

**Author affiliations**
[1]Radcliffe Department of Medicine, University of Oxford, Oxford, UK
[2]Nuffield Department of Primary Care Health Sciences, University of Oxford, Oxford, UK
[3]Emergency Multidisciplinary Unit, Abingdon Hospital, Oxford Health NHS Foundation Trust, Oxford, UK
[4]Department of Geratology, Oxford University Hospitals NHS Foundation Trust, London, UK
[5]Department of Public Health and Primary Care, Imperial College London, London, UK
[6]Centre for Prevention of Stroke and Dementia, Nuffield Department of Clinical Neurosciences, University of Oxford, Oxford, UK
[7]NIHR Oxford Biomedical Research Centre, John Radcliffe Hospital, Oxford, UK
[8]Nuffield Department of Medicine, University of Oxford, Oxford, UK
[9]Institute of Applied Health Research, College of Medical and Dental Sciences, University of Birmingham, Birmingham, UK

**Contributors** CFC undertook analyses and drafted the initial manuscript. GH and TF advised on analytical methodology, interpreted data and critically revised the manuscript. JSTB, TCNE and RH designed data collection methods, collected data, clinically interpreted analyses and critically revised the manuscript. STP

designed the service evaluation, data collection methods and critically revised the manuscript. DSL designed the service evaluation, advised on analytical methods, interpreted data and critically revised the manuscript.

**Funding** CFC is funded by an NIHR Academic Clinical Fellowship. The NIHR Oxford Biomedical Research Centre supported DSL and SP with grant number AC14/035 supporting JSTB and TCNE. JSTB is also supported by the NIHR Oxford Collaboration for Leadership in Applied Health Research and Care (CLAHRC) at Oxford Health NHS Foundation Trust. This article presents independent research funded by the NIHR Community Healthcare MedTech and Invitro Diagnostic Co-operative.

**Disclaimer** The opinions expressed in this study are those of the authors and not necessarily those of the NIHR, the Department of Health or the NHS.

**Competing interests** None declared.

**Patient consent** Detail has been removed from this case description/these case descriptions to ensure anonymity. The editors and reviewers have seen the detailed information available and are satisfied that the information backs up the case the authors are making.

**Ethics approval** This study was part of a service evaluation of the care delivered by the Emergency Multidisciplinary Unit, with routinely collected data used to determine future care design and use of appropriate clinical decision support. The service evaluation was prospectively approved by the Oxford University Hospitals NHS Foundation Trust with Datix reference number 3812. The evaluation was also approved by Oxford Health NHS Foundation Trust Older People's Directorate Audit Department and permission to publish was granted by Oxford Health NHS Foundation Trust Research and Development Department. As this study had approval as a service evaluation, we did not seek formal ethical approval through the national research ethics service.

**Provenance and peer review** Not commissioned; externally peer reviewed.

**Data sharing statement** No additional data are available.

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
