## [Reviewer comments · BMJ Open]

ARTICLE DETAILS

TITLE (PROVISIONAL)	Sepsis recognition tools in acute ambulatory care: associations with process of care and clinical outcomes in a service evaluation of an Emergency Multidisciplinary Unit in Oxfordshire
AUTHORS	Camm, Christian; Hayward, Gail; Elias, Tania; Bowen, Jordan; Hassanzadeh, Roya; Fanshawe, Thomas; Pendlebury, Sarah; Lasserson, Daniel

VERSION 1 – REVIEW

REVIEWER	Dr Tamas Szakmany Cardiff University, UK
REVIEW RETURNED	14-Nov-2017

GENERAL COMMENTS	Thank you for the opportunity to review this well presented paper on the utility of sepsis recognition tools in the ambulatory care setting. The authors described a prospective service evaluation, where they compared the different recognition tools for clinical usefulness. The study could make important contribution to the ongoing debate about the way clinicians should detect sepsis in different environments, however there are several omissions of data which make the results hard to interpret. Specific comments: 1. It is unclear from the Methods section, exactly what variables were collected for the study. This list could be presented in the electronic supplementary material as it becomes important further down the line. Have the researchers used a paper based screening tool? They could provide a copy of this too.2. There is an acknowledgement, that not all of the variables in the NICE-HR and NICE-MR dataset were collected. Is there a systematic omission and if yes, what component(s)? Did the researchers model how would the predictive utility changes if these components were positive? Did they have any systemic omissions in the other scoring systems as well? If yes which parameters were these?3. PPV/NPV sensitivity and specificity is reported, however for the clinical reader presentation of ROC curves could provide a much better visual tool to help to understand the strengths and weaknesses of each scoring system.4. I was surprised that in Table 1 the authors omitted the presentation of NEWS, qSOFA, number of SIRS criteria etc in the two groups. It might even warrant a Figure to describe the distribution of these scores between the two groups, further helping the reader to understand the interconnectedness of these tools.5. It would be particularly interesting to understand why did the authors choose NEWS>4 as their cut-off. In a large analysis of inpatient population by Churpek et al NEWS>7 was found to have
--

	the best predictive value for adverse outcomes [1]. In a recent study our group found that the median NEWS was 4 (3-6) in the sepsis group in patients admitted to the ED or to the normal hospital ward [2]. Many Trusts consider patients with NEWS 3 or above as at risk of adverse outcomes and in Wales this is where escalation of care is first considered [3]. The description of the distribution of the observed NEWS score would be very helpful to understand the results. 6. The results of this study in contrast of the recent Churpek et al. publication would warrant discussion as they presented a similar analysis on a much larger inpatient dataset [1]. 7. The recent change in the definition of sepsis emphasised that its a "bad infection" resulting in organ dysfunction. Not many of the patients in the study would have fulfilled this criteria. On the other hand the sepsis recognition tools were developed for patients who are likely to have "bad infection" and organ dysfunction, so their use could be indeed questionable in a different population. I feel the authors missed an opportunity to look at their clinical setting and perhaps offer other solutions to this problem, based on their data. If the current tools are not sufficient, is their any physiology based tool which could give better results? Minor points: The Introduction is too long and the fifth paragraph describing the ambulatory care model could easily be moved to the Discussion.  1. Churpek MM, Snyder A, Han X, Sokol S, Pettit N, Howell MD, et al. Quick Sepsis-related Organ Failure Assessment, Systemic Inflammatory Response Syndrome, and Early Warning Scores for Detecting Clinical Deterioration in Infected Patients outside the Intensive Care Unit. Am J Respir Crit Care Med. 2017 Apr 1;195(7):906–11. 2. Szakmany T, Lundin RM, Sharif B, Ellis G, Morgan P, Kopczyńska M, et al. Sepsis Prevalence and Outcome on the General Wards and Emergency Departments in Wales: Results of a Multi-Centre, Observational, Point Prevalence Study. PLoS ONE. 2016;11(12):e0167230. 3. Hancock C. A national quality improvement initiative for reducing harm and death from sepsis in Wales. Intensive Crit Care Nurs. Elsevier; 2015 Apr;31(2):100–5.
--	--

REVIEWER	michael smyth University of Warwick, England
REVIEW RETURNED	11-Jan-2018

GENERAL COMMENTS	Thank you for inviting me to review this manuscript. In summary, the manuscript is well written, describes the characteristics of a cohort of patients who presented to the EMU, and thereafter reports on the performance of various screening tools to aid the identification of patients with sepsis who require IV antibiotics/ Iv fluids, or further care. The statistical methods described are appropriate, and findings are presented in a manner that is readily digestible to the non-statistician. The discussion appropriately addresses limitations. The conclusion is reasonable given their findings.
---

	My only criticism of the manuscript, might be the omission of the UK Sepsis Trust Red/Amber flag screening tools from the evaluation. These tools have been in existence for several years and have been adopted in many areas of practice within the UK. Although it is entirely the authors choice as to which screening tools to include for comparison/assessment, it seems odd that such an established tool has been omitted. I respectfully suggest this is a point for the editor to consider.
--	--

VERSION 1 – AUTHOR RESPONSE

Reviewer: 1

Reviewer Name: Dr Tamas Szakmany

Institution and Country: Cardiff University, UK

Please state any competing interests: Patent application No 1616557.3 for "Assay for distinguishing between sepsis and systemic inflammatory response syndrome", filed in the UK

Please leave your comments for the authors below

Thank you for the opportunity to review this well presented paper on the utility of sepsis recognition tools in the ambulatory care setting. The authors described a prospective service evaluation, where they compared the different recognition tools for clinical usefulness.

The study could make important contribution to the ongoing debate about the way clinicians should detect sepsis in different environments, however there are several omissions of data which make the results hard to interpret.

Specific comments:

1. It is unclear from the Methods section, exactly what variables were collected for the study. This list could be presented in the electronic supplementary material as it becomes important further down the line. Have the researchers used a paper based screening tool? They could provide a copy of this too.

RESPONSE: We thank the reviewer for highlighting this issue. Data were collected from multiple sources including clinical clerkings which were undertaken using a standardised proforma, ambulance sheets, GP referral letters, and communication with patients and relatives. We have now clarified this in the Methods section (Process of Care and Clinical Outcomes paragraph) as follows

"Patients' initial assessment was undertaken using a structured clinical clerking proforma including a brief cognitive test (the abbreviated mental test score (AMTS) and delirium screen) as described previously.¹⁸ Data for analyses were extracted from the clerking proforma supplemented by information from ambulance sheets, GP referral letters, and communications with patients and relatives as listed in Supplementary Table 1"

Data was then inserted onto an excel spreadsheet. As a result, there is not a paper-based screening tool. We have now provided a list of variables collected as Supplementary Table 1. The proforma does not add any extra information than the list of supplementary variables which we have now recorded in the Supplementary Table 1 and to avoid repetition we have not included it.

2. There is an acknowledgement, that not all of the variables in the NICE-HR and NICE-MR dataset were collected. Is there a systematic omission and if yes, what component(s)? Did the researchers model how would the predictive utility changes if these components were positive? Did

they have any systemic omissions in the other scoring systems as well? If yes which parameters were these?

RESPONSE: We appreciate the point raised by the reviewer which we had already commented on in the discussion, but agree that this should be clearer in the manuscript. We have undertaken several steps to address this point and improve the clarity of the methods.

Firstly, we have updated Supplementary Table 2 to list the components of the score that were not available from the clinical clerking. The clinical clerking did not include a specific question for the presence or absence of these additional variables used in the NICE-HR and NICE-MR scores. Given that we were reliant on the clinician's recording of, for example, the observation of whether or not mottled skin was absent or present, we felt that it may introduce bias if we inferred that a feature such as 'mottled skin' was not present if it was not specifically commented upon or recorded.

The Sepsis Scoring Systems paragraph now reads

"Within selected scoring systems, (NICE-MR, NICE-HR), some variables are not systematically available at the first assessment in an acute setting (e.g. urine output, usual blood pressure values for calculation of difference from observed value) and Supplementary Table 2 lists these variables together with any additional clinical features that were not systematically sought within the clerking proforma. The scores were calculated without use of these items which would reflect real world practice after implementation."

Our rationale is that some variables in the new NICE scores are not readily and systematically available as 'front door' acute care assessments where patients have been transferred from community settings. Urine output, for example is not monitored in community settings and very often the usual blood pressure for a patient from primary care records would not be available at this initial acute care assessment - patients who have been transferred for an acute care assessment after calling for an ambulance or after being seen by an out of hours GP service would not have a print out of their GP record with these values. Given these constraints and the lack of published data showing agreement between observers over what is, and what is not 'mottled skin', our methods and results would more closely follow the actual implementation of these new NICE scores, reflecting what is available to the first acute care clinician who is assessing a patient that has been transferred from a community setting.

There were no systemic omissions in other scoring systems and this has also been detailed in Supplementary Table 2.

We have not directly modelled the change in predictive utility if the missing NICE components were positive for the following reason. Our analyses confirm very high sensitivity for both NICE MR and HR scores (so the addition of extra features would not improve this measure) but specificity is low for both NICE MR and HR scores. In other words, the NICE MR and HR scores identify patients as having sepsis who do not in fact have serious bacterial infections. If we were to model the addition of features that are subject to variation in clinical observation such as 'mottled skin' then we are likely to drive down specificity even further. Also, it is not clear how we should undertake modelling without empirical data. For example, we do not know the prevalence of mottled skin in sepsis, or where skin appears to be mottled in conditions other than sepsis, and we do not know the inter-observer variability in deciding when mottled skin is either present or absent. The lack of published empirical data in this area implies that we would need to make multiple assumptions for a modelling exercise which we feel would not be a robust methodological approach.

3. PPV/NPV sensitivity and specificity is reported, however for the clinical reader presentation of ROC curves could provide a much better visual tool to help to understand the strengths and weaknesses of each scoring system.

RESPONSE: We agree with the reviewer regarding the potential utility of ROC curves and had considered including them in this manuscript. We have now included ROC curves for analyses undertaken within the whole cohort in the supplementary material but we felt that they would not add clarity to the main manuscript for the following reasons:

a. ROC curves are most useful in settings where the threshold can be altered. However, although this is possible with the NEWS scores, the remaining systems tested in this analysis have a defined threshold and only a small number of potential thresholds (4 for qSOFA, 5 for SIRS) limiting their potential utility

b. We were guided by a systematic review published in BMJ Open (Whiting et al. BMJ Open. 2015;5:e008155.) which highlighted that <1% of doctors are familiar with using ROC curves. As a result of the above two considerations we have not included ROC curves in the main manuscript but have included them in the supplementary material to address the point made by this reviewer.

4. I was surprised that in Table 1 the authors omitted the presentation of NEWS, qSOFA, number of SIRS criteria etc in the two groups. It might even warrant a Figure to describe the distribution of these scores between the two groups, further helping the reader to understand the interconnectedness of these tools.

RESPONSE: We thank this reviewer for suggesting this addition which we agree would enhance the results. Table 1 has been changed to include this information. For completeness and to add further clarity of the potential impact of widespread roll out of NICE HR and MR criteria, the proportion of those positive for NICE-HR and NICE moderate- and high-risk criteria have also been included.

5. It would be particularly interesting to understand why did the authors choose NEWS>4 as their cut-off. In a large analysis of inpatient population by Churpek et al NEWS>7 was found to have the best predictive value for adverse outcomes [1]. In a recent study our group found that the median NEWS was 4 (3-6) in the sepsis group in patients admitted to the ED or to the normal hospital ward [2]. Many Trusts consider patients with NEWS 3 or above as at risk of adverse outcomes and in Wales this is where escalation of care is first considered [3]. The description of the distribution of the observed NEWS score would be very helpful to understand the results.

RESPONSE: We agree with the reviewer that further detail behind the selection of the NEWS cut-off should have been provided. The threshold score of >4 was selected as this is detailed by the Royal College of Physicians as the threshold where patients move from Low to Medium risk (Royal College of Physicians. National Early Warning Score (NEWS) - standardising the assessment of acute-illness severity in the NHS. July 2012, p.15). Further detail has been added to the manuscript to highlight the reasoning behind this selection. We appreciate that, by its nature, the NEWS can have multiple thresholds and these are differentially applied nationally. However, to allow for comparison with other potential sepsis tools we selected the above threshold from national guidance.

The paragraph Sepsis Scoring Systems now includes the sentence

"A cut-off score of >4 was used for NEWS in line with guidance from the Royal College of Physicians (London) which details this threshold for separating low-risk patients from those at increased risk.⁷"

6. The results of this study in contrast of the recent Churpek et al. publication would warrant discussion as they presented a similar analysis on a much larger inpatient dataset [1].

RESPONSE: We thank this reviewer for highlighting this publication to us. We have added discussion of this paper to the discussion of our current manuscript. While we agree that Churpek et al. have provided useful data for the inpatient population, the group we are examining is different in a number of ways. In particular the community setting is quite different from the ED/medical ward setting in terms of prevalence of sepsis. Further, the average age of patients examined by Churpek et al. was around 16 years younger than those in our current cohort. As such we feel that our current manuscript adds significantly to the data regarding the utility of these metrics in the community and for older people at initial acute assessment.

7. The recent change in the definition of sepsis emphasised that its a "bad infection" resulting in organ dysfunction. Not many of the patients in the study would have fulfilled this criteria. On the other hand the sepsis recognition tools were developed for patients who are likely to have "bad infection" and organ dysfunction, so their use could be indeed questionable in a different population. I feel the authors missed an opportunity to look at their clinical setting and perhaps offer other solutions to this problem, based on their data. If the current tools are not sufficient, is there any physiology based tool which could give better results?

RESPONSE: We acknowledge the point raised by the reviewer in this comment and in fact this is precisely the point of our paper. It is important to have sepsis recognition tools at all acute settings in the urgent care pathway but our findings imply that such tools need to be tailored to their clinical context. It is absolutely true that very few of our patients had a "bad infection" as this reviewer puts it, but nevertheless, acute ambulatory care is an acute assessment setting and will undergo significant growth - NHS Improvement has asked acute trusts to see as many as 30% of their acute medical assessments within ambulatory care.

The aim of this analysis was to establish if the commonly available tools are able to provide predictive utility in determining the risk of a patient developing sepsis. The NICE guidelines released in 2016 highlighted the importance of recognising sepsis at 'the front door'. It is with this in mind that we set out to undertake this analysis.

We agree that determining better tools/markers is an important goal in this area. However, while we felt that the current cohort may provide data for preliminary work into the development of a new tool, we feel that the development and validation of such a new tool is beyond the scope of this paper. We hope that the results of the current analysis will highlight the issues with available models in predicting sepsis in the acute ambulatory care unit and in particular in the older adult, and therefore provide the rationale for further studies.

Minor points:

The Introduction is too long and the fifth paragraph describing the ambulatory care model could easily be moved to the Discussion.

RESPONSE: Whilst we appreciate this reviewer's comments, we feel there is a need to explain to the general reader why the acute ambulatory care context is different from other healthcare settings to facilitate interpretation of the paper. The main findings are that existing sepsis recognition tools may vary in their utility in different clinical contexts. As such, the introduction should, in our view, offer a brief overview of why ambulatory care has developed and why it is different to other settings in the urgent care pathway.

1. Churpek MM, Snyder A, Han X, Sokol S, Pettit N, Howell MD, et al. Quick Sepsis-related Organ Failure Assessment, Systemic Inflammatory Response Syndrome, and Early Warning Scores for

- Detecting Clinical Deterioration in Infected Patients outside the Intensive Care Unit. *Am J Respir Crit Care Med.* 2017 Apr 1;195(7):906-11.
2. Szakmany T, Lundin RM, Sharif B, Ellis G, Morgan P, Kopczynska M, et al. Sepsis Prevalence and Outcome on the General Wards and Emergency Departments in Wales: Results of a Multi-Centre, Observational, Point Prevalence Study. *PLoS ONE.* 2016;11(12):e0167230.
3. Hancock C. A national quality improvement initiative for reducing harm and death from sepsis in Wales. *Intensive Crit Care Nurs.* Elsevier; 2015 Apr;31(2):100-5.

RESPONSE: we thank this reviewer for highlighting these references.

Reviewer: 2

Reviewer Name: Michael Smyth

Institution and Country: University of Warwick, England

Please state any competing interests: none declared

Please leave your comments for the authors below

Thank you for inviting me to review this manuscript. In summary, the manuscript is well written, describes the characteristics of a cohort of patients who presented to the EMU, and thereafter reports on the performance of various screening tools to aid the identification of patients with sepsis who require IV antibiotics/ IV fluids, or further care.

The statistical methods described are appropriate, and findings are presented in a manner that is readily digestible to the non-statistician.

The discussion appropriately addresses limitations. The conclusion is reasonable given their findings.

My only criticism of the manuscript, might be the omission of the UK Sepsis Trust Red/Amber flag screening tools from the evaluation. These tools have been in existence for several years and have been adopted in many areas of practice within the UK. Although it is entirely the authors choice as to which screening tools to include for comparison/assessment, it seems odd that such an established tool has been omitted.

I respectfully suggest this is a point for the editor to consider.

RESPONSE: We thank the reviewer for examining our manuscript. We considered inclusion of the UK Sepsis Trust screening tools in the analysis. However, we decided against this for several reasons:

a. The UK Sepsis Trust provides a number of tools that could have been applicable to our cohort (<https://sepsistrust.org/education/clinical-tools/>). In particular, the EMU setting fits somewhere between general practice, prehospital, and acute medical unit care. The decision as to which screen tool to utilise was felt to be largely arbitrary.

b. Both the ED/AMU and Prehospital screen tools make mention of the NEWS score as an initial triage system. This makes analysis of this screening system and comparison with the other established tools in this analysis complex as NEWS is used as a triage tool on its own.

c. The UK Sepsis Trust Red/Amber Flag system was integrated into the new NICE guidelines with the same criteria and thresholds. As a result, the UK Sepsis Trust Amber Flags correspond with the moderate-risk criteria and the Red Flags correspond with the high-risk criteria used within this analysis.

In order to address the point made by this reviewer, we have added the following line to the methods section (Sepsis Scoring Systems)

"We did not separately analyse the Sepsis Trust Red/Amber flag system (<https://sepsistrust.org/education/clinical-tools/>) as features of this system were incorporated into the new NICE guidelines."

Thank you once again for inviting us to respond to the reviewers' comments which have considerably strengthened the manuscript.

Professor Daniel Lasserson, on behalf of the authors.

VERSION 2 – REVIEW

REVIEWER	Dr Tamas Szakmany Cardiff University, UK
REVIEW RETURNED	16-Feb-2018
GENERAL COMMENTS	Thank you very much for addressing all my questions and congratulations on this well written manuscript.
REVIEWER	Michael Smyth University of Warwick, England
REVIEW RETURNED	19-Feb-2018
GENERAL COMMENTS	Many thanks for inviting me to review this revised manuscript. I believe the authors have adequately addressed the concerns raised by the reviewers. I have no further comment.

VERSION 2 – AUTHOR RESPONSE

Dear Hemali

Thank you for inviting us to re-submit our manuscript and we are grateful to the reviewers and thank them for their very positive comments on the re-submission which we note require no additional action.

In order to address the editorial comments, we have changed the title to "Sepsis recognition tools in acute ambulatory care: associations with process of care and clinical outcomes in a service evaluation of an Emergency Multidisciplinary Unit in Oxfordshire" so that the location of the study is now clear.

A STROBE checklist has now been uploaded.

We have clarified the governance with the sentences "This study was part of a service evaluation of the care delivered by the Emergency Multidisciplinary Unit, with routinely collected data used to determine future care design and use of appropriate clinical decision support. The service evaluation

was prospectively approved by the Oxford University Hospitals NHS Foundation Trust with Datix reference number 3812. The evaluation was also approved by Oxford Health NHS Foundation Trust Older People's Directorate Audit Department and permission to publish was granted by Oxford Health NHS Foundation Trust Research and Development Department. As this study had approval as a service evaluation, we did not seek formal ethical approval through the national research ethics service."

Best wishes

Daniel Lasserson, on behalf of the authors.